# Psychological distress in late adolescence: The role of inequalities in family affluence and municipal socioeconomic characteristics in Norway

**Tommy Haugan**[1]*, **Sally Muggleton**[2], **Arnhild Myhr**[3]

**1** Faculty of Nursing and Health Sciences, Nord University, Bodo, Norway, **2** Faculty of Health, University of Canberra, ACT, Bruce, Canberra, Australia, **3** SINTEF Digital, Trondheim, Norway

* tommy.haugan@nord.no

**Data Availability Statement:** The data and materials from the Ungdata-surveys are closed and stored in a national database administered by Norwegian Social Research (NOVA). The present

## Abstract

The present study aims to explore, in the national context of Norway, how municipal socioeconomic indicators affect anxiety and depressive symptom scores among senior high school students and whether this potential municipal effect is dependent on the adolescents' family affluence levels. This cross-sectional study is based on questionnaire data collected in five waves (2014–2018) of the Ungdata survey. The study sample consisted of 97,460 adolescents aged 16–18 years attending high school in 156 municipalities in Norway. Measures of psychological distress, depression, and anxiety symptoms were based on the screening instrument, Hopkins Symptom Checklist-10. Two-level random intercept models were fitted to distinguish the individual and municipality sources of variation in adolescents' mental health. In general, the results indicate substantial psychological symptom load among the study sample. Inequalities in adolescents' psychological distress between family affluence groups were evident, with the lowest symptom loads in the most affluent families. The predicted depressive and anxiety symptoms among the students increased slightly along with the percentage of municipal residents with tertiary educations and with increasing income inequalities in their residential municipality. However, the interaction models suggest that the adverse effects of higher municipal education level and greater income inequality are, to a certain extent, steepest for adolescents with medium family affluence. This study highlights two key findings. Both municipality effects and family affluence account for a relatively small proportion of the total variance in the students' psychological symptoms loads; however, the mental health inequalities we explored between socioeconomic strata on both the individual and municipal levels are not insignificant in a public health perspective. Results are discussed in the context of psychosocial mechanisms related to social comparison and perceptions of social status that may be applicable in egalitarian welfare states such as Norway.

study and analysis of the Ungdata were approved by the Norwegian Centre for Research Data (NSD). Norwegian legislation prohibits deposition of these data to open archives. The data are freely available for research purposes upon application. Details about the application process to NSD can be found at: https://nsd.no/nsddata/serier/ungdata_eng.html.

**Funding:** Funding for open access publication fees was received from Faculty of Nursing and Health Sciences at Nord University.

**Competing interests:** The authors have declared that no competing interests exist.

# Introduction

## Contemporary society and trends in adolescent mental health

The secular trends in adolescent mental health are, in many Western countries, unclear due to changes in recognition, diagnosis, and how adolescents perceive their health [1,2], and this perception appears to be deteriorating among adolescent girls [3]. In Norway, as in other Nordic countries, reports of mental health issues have increased in recent cohorts of adolescents [2,4,5]. The apparent increase in Nordic adolescents affected by depressive symptoms and suffering from mental health disorders must be considered in the context of social and cultural changes that have occurred during the 21st century. Essential aspects of the past year's societal changes could have affected people's mental health status through psychological factors. These modern trends include (I) rising income inequality [6,7], (II) changes in family consultations and dynamics [8,9], and (III) particularly among adolescents, the growing use of modern online technology [10–12] such as social networking sites, which have created a significant arena for social comparisons that could be psychologically harmful [13–15]. Furthermore, secondary and tertiary students commonly report high levels of academic-related stress [16], and feelings of anxiety linked to schoolwork are common, particularly among girls [17], across OECD countries [18]. The social factors are presented at different levels of society: individual, family, community, and national [6].

## Adolescence, social gradient in mental health, and family socioeconomic position (SEP)

The social gradient in mental health, as in health in general, is a widespread phenomenon, and it has been well documented worldwide [19–22]. Socioeconomic inequalities affecting health emerge early on in life and occur across the life course [23,24]. Among adolescents, several studies have demonstrated a higher prevalence of mental health issues with decreasing family SEP calculated using the parents' income, education. and occupational levels [24–26]. Low affluence may affect mental health both directly and indirectly through a variety of mechanisms. Growing up in families with limited socioeconomic resources may affect both access to resources (e.g., time with parents, leisure activities, books and learning materials, housing conditions) and psychosocial conditions among the parents (e.g., stress, conflicts between parents, psychological difficulties,), which together affect the children's socio-emotional development [27]. According to the relative deprivation theory, it is also crucial how adolescents perceive their situations relative to others [28]. Being unable to afford goods and activities that are considered affordable to most can be detrimental to mental health, particularly in adolescence when peer influences are intense [29,30] as the material display of social status—how symbols of wealth are consumed and displayed (i.e., symbolic capital)—may be as important as the income itself [31].

Adolescence, particularly late adolescence [32], may well be the period of life with the highest social equalization in health [33] as adolescents earn independence from their parents but are still in the process of determining their own socioeconomic potential in terms of education, income, and job opportunities [34]. However, several studies demonstrate an association between subjective SEP and adolescent health outcomes, particularly regarding mental health, [35], although the health inequalities are considered less clear and consistent than those for adults [35,36]. Lack of consistent findings may be due to geographical differences [35], methodological factors, use of different measurements of SEP and mental health [37,38], or the possibility that SEP may have different effects on various dimensions of mental health [25] and at different stages of adolescence. Measuring adolescents' SEP is an important methodological

challenge in examining health disparities among young people. School-aged children and youth are likely to still be attending school and living with their parents, and consequently they lack their own social and economic status [26,39]. Material conditions in the family can also be measured by the Family Affluence Scale (FAS) and used as an alternative to uncertain estimates of household income, parental education level, and parents' occupational status/position [40]. Inequalities in self-reported health and psychosomatic symptoms in adolescents have been revealed using FAS [40,41]. However, Amone-P'Olak et al. [25] reported that adolescents in the low SEP category are at risk of mental health problems, but family SEP accounts for a relatively small proportion (≤ 5%) of the total variance in mental health. Another aspect to consider is that objective SEP relates differently to adolescent health than does subjective social status (SSS) [37], and they are not equivalent indicators of the same construct. It may be that SEP differences in adolescent health relate more closely to psychosocial processes than to material inequality [37].

## Socioeconomic environment, psychological distress, and social comparison

Previous studies have emphasized different contextual domains that are likely to affect mental health [42] such as the structure and capacity of health care resources in a community [43,44], local economic conditions [45–47], income inequality [48–50], social disruption [51,52], and social capital [53,54]. Lantz, Pritchard [55] define socioeconomic environment "as a place with geographically defined boundaries that also has economic, educational, social, cultural, and political characteristics," and it can be measured within different units of geography (e.g., census tract, school district, zip code, municipality, county, country). In addition to family-related SEP, the socioeconomic environment shapes resources, opportunities, and exposures that can both directly and indirectly influence health outcomes [55]. For example, in comparison to young people from wealthier neighborhoods, those from areas with high levels of poverty and distress tend to have higher levels of psychological distress and are at higher risk of suffering physical issues; however, neighborhood deprivation is not necessarily a causal risk factor for poor health nor for damaging health behaviors [56,57]. Previous research suggests that people living in more equal communities in terms of material standards report better physical and mental health than do their peers in less equal communities [58].

However, it is not a straightforward process to determine which socioeconomic conditions and community circumstances are considered beneficial for adolescents' mental health and which conditions have an adverse impact [59]. From a social interaction perspective [60], one might expect that adolescents living in high socioeconomic communities achieve health benefits from living in an environment that promotes an active and healthy lifestyle, having access to high-quality health services when needed [61], and surrounded by people with hope in life and faith in the future who facilitate (positive) learning, play, and growth. Moreover, it could be argued that living in a safe neighborhood with high materialistic standards has a positive psychological effect [62]. Alternatively (and worth considering even though the literature is sparse), adolescents residing in high-level socioeconomic communities may be prone to creating harmful stress for themselves as a result of being exposed to social comparison and pressure to achieve. Being surrounded by ambitious and competitive fellow students from well-educated families may, for example, produce high academic stress due to educational expectations and pressure for academic achievement [63,64] as well as difficulties with social comparison making them feel inferior and worthless [65]. Other school-based social status dimensions, such as being attractive and sporty, have also been associated with well-being and psychological distress [66,67], and these might be reinforced in communities with high socioeconomic profiles.

A weakness of many previous studies is that family SEP is not seen in relation to the socio-economic standards of the local community. The potential interplay between the effects of structural, family, and community resources on adolescents' depressive symptoms and well-being may manifest in various ways. For example, in line with the contextual amplification hypothesis [68], we might assume that the detrimental influence of familial factors and adverse community conditions, including concentration of poverty, noxious environment, weak degree of social integration, and collective socialization, reinforce one another. From another perspective, we might assume a buffer effect brought about by the beneficial resources at the family level which protect against adverse influence on the community level [69], or vice versa. However, an alternative explanation may be that the beneficial influence of family social resources on adolescents' depressive symptoms decrease under adverse community conditions. Wickrama, Bryant [70] call this "moderation of a contextual dissipation" and argue that, instead of buffering, the negative influence of family social resources on adolescents' depressive symptoms level off under adverse community conditions.

Several psychosocial factors could increase adolescents' vulnerability to symptoms of depression and anxiety. Perceiving oneself as having low social rank compared to others has been demonstrated to be consistently linked to a higher degree of depressive symptoms [71]. Wetherall et al. [71] stated in their review article that "although markers of SES are consistently associated with depressive symptoms (e.g., [44], measures of social rank may have a stronger association as they tap more psychosocial constructs than the objective indicators (Marmot and Wilkinson, 2001 [20])." Adolescents' perceptions of social status may be two-dimensionally rooted [72]. One dimension is class identification based on parental SEP and familial placement in society: adolescents in high-level socioeconomic communities might live in families with relatively low or moderate socioeconomic resources; these adolescents are particularly vulnerable to the adverse effects of social comparison. Another dimension is the adolescents' sense of personal standing compared to peers/schoolmates when it comes to, among other things, school performance and popularity in their school community. As adolescents age, they undergo a process of cognitive maturation that may increase their self-conception and ability to place themselves on a social status ladder [72]. In Norway, young adults without high school diplomas face a higher risk of receiving a medically based disability pension (where the leading cause is mental illness) before turning 40-years-old if they reside in a municipality with a high socioeconomic profile [73].

## Study aims

As income and educational inequalities in Norway are rising [74], socioeconomic disparities in adolescents' mental health may be rising concurrently. Furthermore, within a community, the average educational level should be considered as a proper proxy measure of the socioeconomic profile of the area, considering local economic conditions and other social community benefits [75,76]. On the other hand, there may be psychological mechanisms associated with well-educated communities such as competitive-oriented environments, higher expectations, and a rush for status that also impact (in a positive or negative way) on adolescents' mental health. Despite this, few studies have explored the interactions between individual-level family affluence and community-level socioeconomic profile. This study aims to explore, in the national context of Norway, how the municipal socioeconomic indicators *education level* and *income inequality* affect anxiety and depressive symptom scores among students in their later teenage years, as well as to understand to what extent these associations are conditioned by family affluence. This study asked the following three research questions: (I) Do high school students (in general) achieve psychological benefits by living in a municipality with a high

average education level? (II) Does the level of income inequality in a municipality affect the psychological symptom load among students? (III) Is this hypothetical municipal effect on high school students dependent on family affluence level?

## Methods

### Design and data sources

**The Ungdata survey.**   The studies involving human participants were reviewed and approved by the Norwegian Centre for Research Data (NSD), https://www.nsd.no/en/. Informed consent to participate in this study was obtained from the students, as well as from legal guardian/next of kin if the student was under 16 years of age. This cross-sectional study is based on questionnaire data collected in five waves (2014–2018) of the Ungdata survey [77,78]. Ungdata is a quality assured and standardized system for local questionnaire surveys aimed at adolescents attending high school in Norway. The Welfare Research Institute NOVA (at Oslo-Met) is, together with Norway's seven regional drug and alcohol competence centers (KoRus), responsible for conducting the survey.

Participation in the survey is voluntary and based on the students' informed consent. The survey covers different aspects of the students' lives encompassing a wide range of thematic areas, and it is an important source of information on young people's health and well-being, both at the municipal and national levels. The surveys take place during school hours and are carried out electronically. The response rate varies between surveys, schools, and school years. The overall response rate among senior high school students was 66% for surveys conducted in 2014–2016 and 69% for surveys conducted in 2016–2018 [77–79]. According to Bakken [77], the data are considered nationally representative during a three-year period. Our data are based on participants in the Ungdata survey during the period 2014 to 2018, and we thus consider that our study material in general gives a representative picture of Norwegian youths. (See Frøyland [80] for a detailed description of the content and theoretical framework of the Ungdata survey.)

**Municipality state reporting (KOSTRA).**   We combined the individual data with census information on individuals' home municipalities, sourced from Municipality-State-Reporting (KOSTRA) database administered by Statistics Norway [81] (SSB). KOSTRA is a national information system that contains management information on municipal key activities including, demography, economy and social services development.

### Study population

Our study sample consists of students who voluntarily undergo three years of education at level 3 in the International Standard Classification of Education (ISCED), which is the final stage of general and vocational secondary education [82]. In Norway, students generally begin ISCED level 3 at age 16 and complete it the year they turn nineteen. Programs classified at ISCED level 3 may, for example, be referred to as "upper secondary education" or "(senior) high school"; in the present paper, we use the term "high school."

In this study, all high school students completing the version of the Ungdata survey containing questions on psychological distress, anxiety, and depression in the period 2014–2018 were included (N = 144,239). We excluded individuals with missing information on gender (n = 6,072), school year (n = 955), family affluence (n = 687), and municipal residential identifiers (n = 31,923). The final dataset with complete municipal identifiers contained 104,602 individuals. The study sample was further reduced in the parametric estimations due to individuals' missing information on mean scores of psychological distress (n = 8,095), symptoms of depression (n = 7,142), and symptoms of anxiety (n = 7,459). We found no statistical

difference between included and excluded individuals for mean symptom scores of psychological distress (1.91 vs. 1.91), depression (2.16 vs. 2.15), or anxiety (1.55 vs. 1.55). There was a noted higher share of first-year students (54% versus 45%, p<0.001) for excluded compared to included groups.

The fact that first-year high school students are overrepresented in the study population should be mentioned and may partly be explained by the fact that some high schools only offered the youngest students the opportunity to participate in the surveys [83]. Moreover, it is more difficult to carry out surveys among the oldest students due to exams and the fact that many of the students are apprentices. A highly relevant explanation is also that the dropout rate in high school increases with increasing age of students. We should therefore be aware that the data, especially towards the end of high school, consists of a more selected group of students compared to lower grade levels.

### Assessment of variables

**Psychological distress, and depressive and anxiety symptoms.** Our dependent variables are adolescent's psychological distress, and depressive and anxiety symptoms. Psychological distress was measured using the 10-item Hopkins Symptom Checklist, consisting of two sub-scales: a depression dimension (six items that constitute the "Depressive Mood Inventory") and an anxiety dimension (four items). In addition, there is a total mean score (10 items) [84,85].

The students reported how often they were bothered by each of the following symptoms during the past week (the six first items relate to the depression dimension of the scale): "felt that everything is a struggle"; "had sleep problems"; "felt unhappy, sad, or depressed"; "felt hopeless about the future"; "felt stiff or tense"; "worried too much about things"; "suddenly felt scared for no reason"; "felt constant fear or anxiety"; "been nervous or felt uneasy"; and "felt worthless". Each item was answered on a four-point scale ranging from "not at all" (1) to "very much" (4).

Separate measures for psychological distress (total mean) and depressive/anxiety symptoms were constructed by adding up the scores (1 to 4) on all the items covering each dimension (10 items in total: 6 for depression and 4 for anxiety) and dividing the total by the number of completed items, given responses to at least half the statements for each scale. The resulting mean symptom scale scores—one for psychological distress, one for symptoms of depression, and one for symptoms of anxiety—ranged from 1 to 4. In addition to the mean score, a validated cut-off score of ≥1.85 was used to identify students reporting moderate to high symptom loads related to overall psychological distress, depression, and anxiety [84].

**Socioeconomic position (SEP).** The socioeconomic position (SEP) of the adolescents was measured using a collective measure of SEP developed by Bakken et al. [83] which includes, in addition to four questions from FAS II [39,40], information on parental education levels and the number of books in the home. The students answered the following four questions retrieved from FAS II: "Does your family have a car?"; "Do you have your own bedroom?"; "How many times have you travelled somewhere on holiday with your family over the past year?"; and "How many computers or tablet computers does your family have?" FAS II has been validated alongside other measures of adolescents' SEP and compared to measures in which adolescents report their parents' income, occupations, and education levels, and the scale has been found to have better criterion validity and less susceptibility to non-response bias [86].

The goal with the collective measure of SEP is to capture three dimensions of a family's socioeconomic position—parents' education level, number of books in the home, and the

family's affluence level—and combine these into a collective measure of the family's socioeconomic status. Parents' level of education, number of books in the home, and family's affluence level each have some clear limitations as measures of a family's socio-economic situation. A collective index based on these three dimensions will probably provide a more robust and valid measure of family socioeconomic status [87]. Consideration of anonymity is the main reason that the Ungdata surveys do not include questions about parents' occupations or incomes. A critical review of each question included in the collective affluence measure, as well as detailed information on how the measure is developed, may be found in Bakken et al. [83]. We calculated a mean sum score, ranging from 0 to 3, for each study participant, and then the total study sample was split into three equally sized groups ordered by increasing affluence level from low to high (low, medium, and high).

**Municipal sociodemographic characteristics.** In our descriptive analysis, we included municipal education level, median gross household income, income inequality, unemployment rate, disability pension rate, immigration rate, divorce rate, and life expectancy. The municipal education level was defined as the percentage of municipal inhabitants aged 15–80 years who completed tertiary education. In the descriptive analyses, the municipalities were grouped into quartiles where Quartile 1 is the 25% of municipalities with the lowest education level; Quartile 2 is the second lowest education level group; Quartile 3 is the group of municipalities with the second highest education level, and Quartile 4 is the 25% of municipalities with the highest education level among the adult population.

The Gini coefficient, the most commonly used measure of income inequality [88,89], was used to estimate income inequality within municipalities. We used the census Gini coefficient (only available in the period 2014–2017) calculated for each municipality by Statistic Norway. The Gini coefficient varies between 0 and 1, where 0 corresponds to so-called "perfect income equality," meaning that every household has the same income and wealth, and 1 corresponds to perfect income inequality, meaning that one household possesses the population's entire income and wealth. According to the OECD, in 2017 Norway was ranked as the sixth country with the least income inequality, with a Gini coefficient of 0.262 [89]. Statistics Norway excludes students and uses a different method when adjusting for large households in their calculations of the Gini coefficient, which in 2017 was 0.252 [88]. The municipality characteristics enter the analyses as continuous (percentages or median) census variables. In our parametric estimations, we only included municipal education level and income inequality.

**Covariates.** The individual variables in Ungdata used in this study are *gender*, *school year* (proxy for age and categorized as follows: first, second, or third year of high school), and *survey cycle* (survey conducted in 2014, 2015, 2016, 2017, or 2018). These variables were included because earlier studies show strong associations with adolescents' mental health [5,77,90,91]. Introducing other individual or contextual sociodemographic variables was considered problematic in the modelling in regards to problems with potential statistical over-adjustment due to uncertainty whether the covariates represent real cofounding issues; or rather act as mediators, colliders or have bi-directionally roles [92].

## Statistical approach

First, descriptive analyses of percentages on individual variables and municipal sociodemographic variables, by municipal education level were tested by chi square tests and analysis of variance (ANOVA). Second, we investigated the relevance of the residential context as well as the association between family affluence, municipal education level/income inequality, and psychological distress, depression, and anxiety symptoms among high school students and tested the hypothetical interactions using multilevel models [93,94].

Linear multilevel models [93–95] with individuals (level 1, n = 97,460) nested within municipalities (level 2, n = 156) were estimated. A two-level random intercept model was fitted using maximum likelihood estimation to distinguish the individual and municipality sources of variation in adolescents' mental health [96]. We modelled the prediction of adolescents' mental health in five steps. First, we estimated an intercept model, only including the random intercept, to determine the impact of the municipality context on adolescents' mental health. Second, we included the individual and family level variables (gender and school year) and family affluence (Table 2). Third, as shown in Table 3, we extended the random intercept model for the relationship between family affluence and adolescents' mental health to allow family affluence effect to vary across municipalities. A likelihood ratio test (LR test) was used to compare the random intercept and the random slope models' goodness of fit. In the final steps (Table 4), we included the municipal education level (main effect model) and the interaction terms of family socioeconomic status with municipal education level (interaction model) for all three mental health outcomes: psychological distress, depressive symptoms, and anxiety symptoms. Similarly, in Table 5 municipal income inequality as well as the interaction terms of family affluence with municipal income inequality were included. In addition, supplementary analyses exploring the association between municipal socioeconomic conditions and adolescent risk of moderate-to-high psychological symptoms (main effect model) as well as analysis of the hypothetical interactions with family affluence (interaction model) were performed; the results are presented in S1–S3 Tables.

Estimates for fixed effects are reported as coefficients with 95% confidence intervals (CI). To quantify the influence of municipality of residence on adolescents' mental health, we computed the intraclass correlation coefficients (ICCs) [96] for each outcome. The ICC expresses the correlation in the outcomes (i.e., psychological distress, depressive and anxiety symptoms) between two individuals randomly selected from the same municipality: the larger the ICC, the stronger the clustering of psychological distress within the municipality and the larger the general contextual effect of the municipality. The Akaike Information Criterion (AIC) and the Bayesian Information Criterion (BIC) were used as measures of goodness of fit for our models. The model parameters were estimated by a general linear model (GLM) mixed effects method using Stata/MP software (version 13).

# Results

## Characteristics of the study population

Table 1 presents descriptive information regarding the individual and municipality contextual variables among high school students in Norway, both in total and by municipal education level. In all, 50.4% females and 49.6% males participated in this study. More first-year students (44.6%) compared to second- (33.7%) and third-year (21.7%) students completed the questionnaire. Each family affluence group was (approximately) equally represented in the study population, with about one-third of the students each in high (36.3%), medium (30.8%), and low (32.9%) affluence groups. The mean family affluence score was 1.9, and, as expected, mean affluence score as well as share of high affluence students increased with increasing municipal education (p<0.001).

Inequalities in adolescents' mental health were observed both between family affluence groups and municipal education groups in all three mental health domains, with increasing symptom load associated with decreasing affluence and education level. Mean psychological distress, depressive, and anxiety symptoms were higher (p<0.001) in low affluence adolescents (1.95, 2.19, 1.58, respectively) compared with their peers with medium (1.91, 2.16, 1.54, respectively) and high (1.87, 2.11, 1.51, respectively) affluence levels (results not shown in table). The

**Table 1. Descriptive characteristics on the variables (by n or mean) used in the present study by four groups of municipal education level.**

| | 25% lowest education level | | Lower middle education level | | Upper middle education level | | 25% highest education level | | Total mean | | p-value[*] |
|---|---|---|---|---|---|---|---|---|---|---|---|
| | n/mean | %/SD | n/mean | %/SD | n/mean | %/SD | n/mean | %/SD | n/mean | %/SD | |
| **Individual and family level variables** | | | | | | | | | | | |
| Female | 12,725 | 49.2 | 12,802 | 50.4 | 12,526 | 50.4 | 14713 | 51.6 | 52,766 | 50.4 | P<0.001 |
| School year | | | | | | | | | | | |
| year 1 | 11,481 | 44.4 | 10,741 | 42.3 | 12,889 | 51.9 | 11562 | 40.6 | 46,673 | 44.6 | P<0.001 |
| year 2 | 8,844 | 34.2 | 9,192 | 36.2 | 8,110 | 32.7 | 9088 | 31.9 | 35,234 | 33.7 | P<0.001 |
| year 3 | 5,545 | 21.4 | 5,475 | 21.5 | 3,836 | 15.4 | 7839 | 27.5 | 22,695 | 21.7 | P<0.001 |
| Mean family affluence | 1.8 | 0.6 | 1.9 | 0.6 | 2.0 | 0.6 | 2.0 | 0.7 | 1.9 | 0.6 | P<0.001 |
| Psychological distress[a] | 1.8 | 0.7 | 1.9 | 0.7 | 1.9 | 0.7 | 2.0 | 0.7 | 1.9 | 0.7 | P<0.001 |
| Depressive symptoms[b] | 2.1 | 0.8 | 2.2 | 0.8 | 2.2 | 0.8 | 2.2 | 0.8 | 2.2 | 0.8 | P<0.001 |
| Anxiety symptoms[c] | 1.5 | 0.6 | 1.5 | 0.6 | 1.6 | 0.6 | 1.6 | 0.6 | 1.5 | 0.6 | P<0.001 |
| *Prevalences moderate to high symptom load* | | | | | | | | | | | |
| Psychological distress | 10,322 | 42.9 | 11,029 | 46.8 | 11,000 | 47.5 | 13,161 | 51.2 | 45,512 | 47.2 | P<0.001 |
| Depressive symptoms | 12,766 | 52.6 | 13,544 | 56.9 | 13,357 | 57.3 | 15,740 | 60.5 | 55,407 | 56.9 | P<0.001 |
| Anxiety symptoms | 5,495 | 22.7 | 5,870 | 24.7 | 5,879 | 25.3 | 7,353 | 28.3 | 24,597 | 25.3 | P<0.001 |
| **Municipality level variables** | | | | | | | | | | | |
| Population | 13842.9 | 12248.5 | 28501.2 | 21008.8 | 62494.2 | 56675.6 | 528217.1 | 243236.1 | 169047.4 | 256137.8 | p<0.001 |
| Median household income | 611788.4 | 53639.8 | 631682.5 | 58009 | 679992.5 | 71945.4 | 648474.6 | 101384.6 | 642805.7 | 78500.5 | p<0.001 |
| Gini-coefficient | 0.22 | 0.02 | 0.23 | 0.01 | 0.25 | 0.02 | 0.33 | 0.01 | 0.26 | 0.05 | p<0.001 |
| % tertiary education | 21.7 | 1.8 | 28.4 | 1.9 | 37.9 | 4.6 | 50.5 | 1.3 | 35 | 11.4 | p<0.001 |
| Unemployment rate | 1.6 | 0.6 | 1.8 | 0.6 | 1.8 | 0.9 | 1.8 | 0.3 | 1.8 | 0.6 | p<0.001 |
| Disability pension rate | 11.8 | 2.5 | 10.7 | 2.5 | 7.6 | 1.5 | 5.2 | 0.3 | 8.7 | 3.2 | p<0.001 |
| Immigration rate | 11.3 | 4 | 14 | 5.2 | 17.2 | 5.1 | 29.8 | 5.4 | 18.4 | 8.8 | p<0.001 |
| Election turnout (%) | 58.1 | 4.4 | 58.5 | 4.1 | 59.3 | 3.2 | 63.2 | 0.3 | 59.9 | 3.9 | p<0.001 |
| Reported crimes (per 1000 inhabitants)[d] | 45.2 | 14.9 | 63.0 | 23.0 | 57.5 | 15.6 | 84.1 | 23.2 | 63.8 | 24.7 | p<0.001 |
| Divorce rate | 11 | 1.9 | 11.2 | 1.7 | 10.3 | 1.3 | 9.9 | 0.3 | 10.6 | 1.5 | p<0.001 |
| Life expectancy | 81.2 | 0.8 | 81.2 | 0.8 | 81.8 | 0.6 | 81.2 | 0.8 | 81.4 | 0.8 | |

[*]Diff. between groups tested by chi-square test or Analysis of variance (ANOVA). Sample size are reduced

[a]n = 96,507

[b]n = 97,460

[c]n = 97,143

[d]n = 95,750.

prevalence of students with moderate-to-high symptom loads showed similar patterns. Furthermore, mean psychological distress (1.8 to 2.0, p<0.001), depressive (2.1 to 2.2, p<0.001), and anxiety (1.5 to 1.6, p<0.001) symptoms increased with increasing municipal education level. Similarly, the prevalence of students with moderate-to-high symptom loads increased with increasing municipal education level in all three mental health domains (psychological distress: 43% to 51%, depressive symptoms: 53% to 61%, and anxiety symptoms: 23% to 28%).

We noted differences in other municipal characteristics across municipal education levels when divided into quartiles (Table 1). The mean share of municipal inhabitants who have completed tertiary education increased from 22% in municipalities with the lowest education levels to 51% in the most educated municipalities. Income inequality (measured by the Gini coefficient) increased incrementally from the 25% least educated municipalities up to the municipal group in the top 25% education level. The most educated municipality (75th percentile) had a larger population (528,217 vs. 169,047 inhabitants), higher median household

**Table 2. The impact of individual characteristics and family affluence on psychological, depressive and anxiety symptoms.**

| | Psychological distress (n = 96,507) | Depressive symptoms (n = 97,460) | Anxiety symptoms (n = 97,143) |
|---|---|---|---|
| | Coef (95% CI) | Coef (95% CI) | Coef (95% CI) |
| **Fixed components** | | | |
| Constant | 1.54 (1.52–1.56) | 1.75 (1.73–1.77) | 1.24 (1.22–1.25) |
| **Individual level** | | | |
| Gender (female) | 0.49 (0.48–0.51) | 0.54 (0.53–0.56) | 0.42 (0.41–0.43) |
| Family affluence | | | |
| High | ref | Ref | Ref |
| Medium | 0.06 (0.05–0.07) | 0.07 (0.06–0.08) | 0.05 (0.04–0.06) |
| Low | 0.09 (0.08–0.10) | 0.09 (0.08–0.11) | 0.08 (0.07–0.09) |
| School year | | | |
| Year 1 | Ref | Ref | Ref |
| Year 2 | 0.01 (-0.003 to 0.02) | 0.01 (-0.002 to 0.03) | 0.004 (-0.01 to 0.02) |
| Year 3 | 0.10 (0.09 to 0.12) | 0.14 (0.12 to 0.16) | 0.04 (0.03 to 0.06) |
| Gender x school year | | | |
| Female x year 1 | Ref | Ref | Ref |
| Female x year 2 | -0.02 (-0.04 to -0.01) | -0.01 (-0.03 to 0.01) | -0.05 (-0.06 to -0.03) |
| Female x year 3 | -0.08 (-0.10 to -0.06) | -0.07 (-0.09 to -0.04) | -0.09 (-0.11 to -0.07) |
| Survey cycle | 0.03 (0.03–0.04) | 0.03 (0.03–0.04) | 0.036 (0.03–0.04) |
| **Random effects** | | | |
| Municipality variance | 0.004 (0.003–0.006) | 0.007 (0.005–0.01) | 0.002 (0.001–0.003) |
| Individual variance | 0.40 (0.396–0.404) | 0.56 (0.56–0.57) | 0.36 (0.35–0.36) |
| ICC (%) | 1.1 | 1.3 | 0.53 |
| AIC | 185706.8 | 220586 | 175784.9 |
| BIC | 185811 | 220690.4 | 175889.2 |

income (648,476 NOK vs. 642,806 NOK), a lower disability pension rate (5.2% vs. 8.7%), a higher immigration rate (30% vs. 18.4%), and a slightly lower divorce rate (10% vs. 10.6%) compared to the average of the municipalities. The unemployment rate was 0.2 lower in the municipal group with the 25% lowest education level of the other groups (1.6 vs. 1.8). The life expectancy was 0.6 years higher in the municipal group "upper middle education level" than in the other groups (81.8 years vs. 81.2 years).

## Individual- and municipal-level variation in adolescents' psychological distress

In the first step of our parametric estimations, an intercept model containing only the second random intercept was estimated. We found that the ICCs for psychological distress, depressive symptoms, and anxiety symptoms among adolescents were 0.017, 0.019, and 0.010, respectively. In other words, our estimations suggest that about 2% of the variation in the students' psychological distress and depressive symptoms could be attributed to differences between municipalities. However, only 1% of the variation in students' anxiety symptoms could be attributed to differences between municipalities.

Table 2 shows the individual and family covariates of the three mental health outcomes: psychological distress (Model 1), depressive symptoms (Model 2), and anxiety symptoms (Model 3). Family affluence was negatively associated with each of the outcomes, with decreasing symptom scores accompanying increasing affluence levels. Being female and a third-year student was correlated with higher psychological symptoms. The interaction terms with gender and school year are negative and statistically significant, indicating that the positive

association between females and psychological distress and depressive/anxiety symptoms decreases over time.

In Table 3, we extend the random intercept models for the relationship between family affluence and a) psychological distress, b) depressive symptoms, and c) anxiety symptoms to allow the impact of family affluence to vary across municipalities. The two-level random

**Table 3. Parameter estimates and log-likelihood values for the random intercept and random slope linear regression models.**

| Parameter | Random intercept | | Random slope (coefficient) | |
|---|---|---|---|---|
| | Coef (95% CI) | SE | Coef (95% CI) | SE |
| **Psychological distress[a] (n = 96,507)** | | | | |
| **Individual level** | | | | |
| Intercept | 1.78 (1.76–1.80) | 0.010 | 1.79 (1.77–1.81) | 0.010 |
| Family affluence | 0.05 (0.04–0.05) | 0.004 | 0.04 (0.03–0.05) | 0.004 |
| **Municipality level random part** | | | | |
| Residual variance intercept | 0.0083 (0.006–0.011) | 0.001 | 0.0068(0.005–0.010) | 0.0012 |
| Residual variance slope | 0.4570(0.453–0.461) | 0.002 | 0.4567(0.453–0.461) | 0.0021 |
| Intercept-slope covariance | | | 0.0004(0.0002–0.008) | 0.0002 |
| -2Log likelihood | 198593.332 | | 198579.96 | |
| BIC | 198639.2 | | 198637.3 | |
| AIC | 198601.3 | | 198590 | |
| ICC (%) | 1.5 | | 1.8 | |
| **Depressive symptoms[b] (n = 97,460)** | | | | |
| **Individual level** | | | | |
| Intercept | 2.01 (1.99–2.04) | 0.012 | 2.02 (2.00–2.04) | 0.012 |
| Family affluence | 0.05 (0.04–0.05) | 0.003 | 0.04 (0.04–0.05) | 0.004 |
| **Municipality level random part** | | | | |
| Residual variance intercept | 0.0125 (0.010–0.017) | 0.002 | 0.0005(0.0002–0.001) | 0.0002 |
| Residual variance slope | 0.6332 (0.628–0.638) | 0.003 | 0.633 (0.627–0.639) | 0.0029 |
| Intercept-slope covariance | | | 0.0104 (0.007–0.015) | 0.0019 |
| -2Log likelihood | 232355.08 | | 232341.14 | |
| BIC | 232401 | | 232398.6 | |
| AIC | 232363.1 | | 232351.1 | |
| ICC (%) | 1.9 | | 1.6 | |
| **Anxiety symptoms[c] (n = 97,143)** | | | | |
| **Individual level** | | | | |
| Intercept | 1.44 (1.42–1.45) | 0.008 | 1.44 (1.43–1.46) | 0.009 |
| Family affluence | 0.04 (0.04–0.05) | 0.003 | 0.04 (0.03–0.04) | 0.003 |
| **Municipality level random part** | | | | |
| Residual variance intercept | 0.004(0.003–0.0056) | 0.0007 | 0.0035(0.002–0.005) | 0.0007 |
| Residual variance slope | 0.396(0.392–0.3994) | 0.002 | 0.396 (0.392–0.399) | 0.002 |
| Intercept-slope covariance | | | 0.0002(0.0001–0.001) | 0.0001 |
| -2Log likelihood | 185884.358 | | 185880.084 | |
| BIC | 185930.3 | | 185937.5 | |
| AIC | 185892.4 | | 185890.1 | |
| ICC (%) | 1.0 | | 0.9 | |

Likelihood ratio test
[a]LR chi2 = 13.37, p-value = 0.0003
[b]LR chi2 = 13.95, p-value = 0.0002
[c]LR chi2 = 4.28, p-value = 0.0387.

**Table 4. The impact of family affluence, municipal education level and their interactions\* on psychological distress, depressive and anxiety symptom scores in high school students in Norway.**

| | Psychological distress | | Depressive symptoms | | Anxiety symptoms | |
|---|---|---|---|---|---|---|
| | Main effect model | Interaction model | Main effect model | Interaction model | Main effect model | Interaction model |
| | Coef (95% CI) | Coef (95% CI) | Coef (95% CI) | Coef(95% CI) | Coef(95% CI) | Coef(95% CI) |
| **Fixed effects** | | | | | | |
| **Individual level** | | | | | | |
| Family affluence | | | | | | |
| High | Ref | Ref | Ref | Ref | Ref | Ref |
| Medium | 0.059 (0.049–0.069) | 0.003 (-0.03–0.036) | 0.067 (0.056–0.079) | 0.008 (-0.031–0.047) | 0.046 (0.037–0.056) | -0.0003 (-0.031–0.030) |
| Low | 0.091 (0.081–0.101) | 0.061 (0.029–0.094) | 0.095 (0.083–0.106) | 0.066 (0.027–0.104) | 0.081 (0.072–0.091) | 0.059 (0.029–0.090) |
| **Municipal level** | | | | | | |
| % tertiary education | 0.004 (0.003–0.006) | 0.004 (0.002–0.005) | 0.005 (0.003–0.007) | 0.004 (0.002–0.006) | 0.003 (0.002–0.004) | 0.002 (0.001–0.003) |
| **Cross-level interactions** | | | | | | |
| Family affluence x % tertiary education | | | | | | |
| High x education | | Ref | | Ref | | Ref |
| Medium x education | | 0.002 (0.001–0.003) | | 0.002 (0.001–0.003) | | 0.001 (0.001–0.002) |
| Low x education | | 0.001 (-0.0001–0.002) | | 0.001 (-0.0002–0.002) | | 0.001 (-0.0002–0.001) |
| **Random effects** | | | | | | |
| Individual variance | 0.400 (0.396–0.404) | 0.400 (0.396–0.404) | 0.561 (0.556–0.566) | 0.561 (0.556–0.566) | 0.357 (0.354–0.360) | 0.357 (0.354–0.360) |
| Municipality variance | 0.003 (0.002–0.005) | 0.003 (0.002–0.005) | 0.006 (0.004–0.008) | 0.006 (0.004–0.008) | 0.001 (0.001–0.002) | 0.001 (0.001–0.002) |
| ICC (%) | 0.8 | 0.8 | 1.0 | 1.0 | 0.4 | 0.4 |
| AIC | 185681.6 | 185673.2 | 220562.3 | 220556.3 | 175763.3 | 175757.5 |
| BIC | 185795.3 | 185805.9 | 220676.1 | 220689.1 | 175877.1 | 175890.3 |

\*Adjusted for respondents' gender and school year.

intercept model, which is nested in the random slope model, is rejected at the 5% significance level (using a likelihood ratio test), suggesting that the impact of family affluence on adolescents' anxiety and depressive symptoms does vary between municipalities.

In relation to the municipal socioeconomic variables (Table 4), we found that municipal education level is associated with higher psychological distress and depressive/anxiety symptoms among high school students in Norway. However, the interaction models suggest that these associations are also conditioned by the family affluence level of the students (Table 4 and Fig 1). Fig 1 demonstrates that the predicted depressive and anxiety symptoms increase with increasing municipal education level among all high school students. Notably, among adolescents at the medium affluence level, the adverse effect of high municipal education level is statistically steeper than for students with high and low family affluence levels. For example, for students that live in medium affluence families, the predictive depressive symptom score is 2.12 in a municipality where 26% of the inhabitants have completed a tertiary degree, compared to 2.24 in municipalities with a tertiary education level of 46%.

The same impact of municipal education and family affluence and their interactions were found on the predicted probability of moderate-to-high depressive and anxiety symptom complaints (S2 Table and S1 Fig). Students from families of medium affluence who lived in one of the Norwegian municipalities with the lowest percentage of residents with tertiary education (for example, 22%) had a 53% chance of experiencing depressive symptoms. Students from families of medium affluence living in municipalities where the tertiary education level was (as high as) 50% had a predicted likelihood of 63% for moderate-to-high depressive symptom complaints.

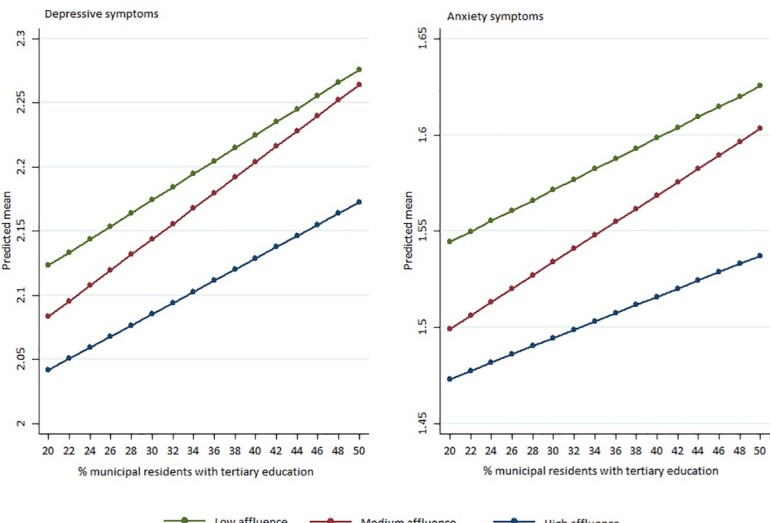

**Fig 1. Predictive margins of high, medium and low affluence high school students predicting mean depressive and anxiety symptoms by percentages of municipal residents with tertiary education.**

The effects of municipal income inequality and its interaction with family affluence on psychological symptoms were also tested, with similar results as with municipal education level (Table 5 and Fig 2, and S3 Table and S2 Fig). Table 5 (main effect models) shows that

**Table 5. The impact of family affluence, municipal income inequality and their interactions\* on psychological distress, depressive and anxiety symptom scores in high school students in Norway.**

| | Psychological distress | | Depressive symptoms | | Anxiety symptoms | |
|---|---|---|---|---|---|---|
| | Main effect model | Interaction model | Main effect model | Interaction model | Main effect model | Interaction model |
| | Coef (95% CI) | Coef (95% CI) | Coef (95% CI) | Coef(95% CI) | Coef(95% CI) | Coef(95% CI) |
| **Fixed effects** | | | | | | |
| **Individual level** | | | | | | |
| Family affluence | | | | | | |
| High | Ref | Ref | Ref | Ref | Ref | Ref |
| Medium | 0.059 (0.049–0.068) | -0.016 (-0.071 to 0.039) | 0.067 (0.055–0.078) | -0.011 (-0.076 to 0.054) | 0.046 (0.037–0.055) | -0.022 (-0.074 to 0.029) |
| Low | 0.090 (0.080–0.100) | 0.072 (0.017–0.126) | 0.093 (0.082–0.105) | 0.084 (0.020–0.148) | 0.081 (0.071–0.090) | 0.062 (0.011–0.113) |
| **Municipal level** | | | | | | |
| Income inequality | 1.062 (0.673–1.451) | 0.952 (0.546–1.357) | 1.188 (0.707–1.669) | 1.088 (0.589–1.588) | 0.813 (0.521–1.105) | 0.708 (0.397–1.020) |
| **Cross-level interactions** | | | | | | |
| Family affluence x income inequality (gini) | | | | | | |
| High x gini | | Ref | | Ref | | Ref |
| Medium x gini | | 0.295 (0.083–0.507) | | 0.304 (0.055–0.554) | | 0.268 (0.069–0.468) |
| Low x gini | | 0.066 (-0.142 to 0.274) | | 0.032 (-0.212 to 0.277) | | 0.067 (-0.128 to 0.262) |
| **Random effects** | | | | | | |
| Individual variance | 0.400 (0.396–0.404) | 0.400 (0.340–0.404) | 0.561 (0.556–0.566) | 0.561 (0.556–0.566) | 0.357 (0.354–0.360) | 0.357 (0.354–0.360) |
| Municipality variance | 0.004 (0.002–0.005) | 0.004 (0.003–0.005) | 0.006 (0.004–0.008) | 0.006 (0.004–0.008) | 0.002 (0.001–0.002) | 0.002 (0.001–0.002) |
| ICC (%) | 0.9 | 0.9 | 1.1 | 1.1 | 0.4 | 0.4 |
| AIC | 185681.7 | 185677.8 | 220565.8 | 220563.3 | 175758.6 | 175755.4 |
| BIC | 185795.4 | 185810.5 | 220679.6 | 220696.1 | 175872.4 | 175888.2 |

\*Adjusted for respondents' gender and school year.

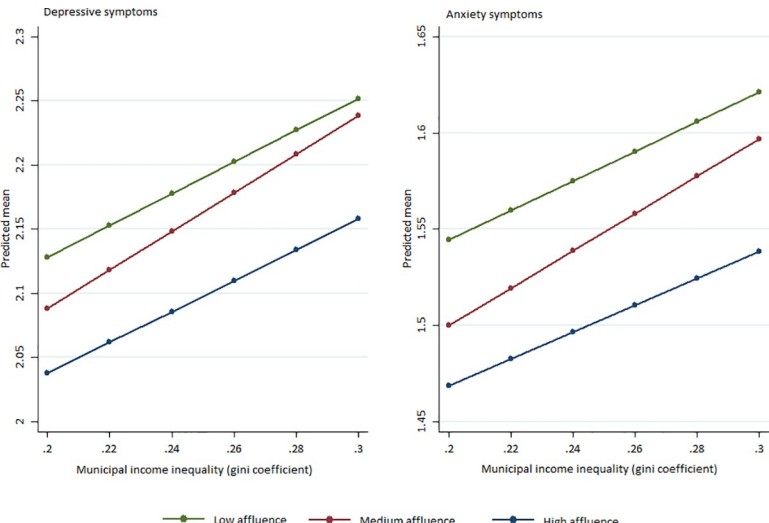

**Fig 2. Predictive margins of high, medium and low affluence high school students predicting mean depressive and anxiety symptoms by municipal income inequality (Gini coefficient).**

psychological distress and depressive/anxiety symptoms among high school students increase with increasing income inequality. Notably, when adding the interaction terms with family affluence, the associations with municipal income inequality and psychological symptom load is most significant for high school students with medium affluence levels (See Table 5 and Fig 2).

## Discussion

### Key findings

This national representative study exploring the impact of interactions between family affluence and municipal socioeconomic characteristics on psychological distress among Norwegian high school students has produced findings that should be highlighted. Overall, our results indicate substantial psychological symptom loads among 16–18-year-old students in Norway. Inequalities in adolescents' mental health between family affluence groups were evident, with increasing symptom loads accompanying decreasing affluence levels. Females, particularly those attending the third year of high school, showed higher psychological symptom scores than males, mostly evidencing moderate-to-high psychological symptom loads. The average psychological symptom load and prevalence of students suffering from moderate-to-high symptom loads increased slightly with increasing education levels and income inequalities in their residential municipalities; this applies to adolescents in all three family affluence groups: low, medium, and high. However, our parametric estimations suggest that these municipal socioeconomic characteristics appear to have the highest impact on the mental health of adolescents living in families at the medium affluence level. This study deepens our understanding of how family affluence interacts with residential contexts to promote or inhibit young people's emotional development.

### Education level in municipality and other socioeconomic conditions

The first research question this study aimed to answer was whether high school students in general achieve psychological benefits by living in a municipality with a high average education

level. Our parametric estimations suggest that the predicted mean of depressive and anxiety symptoms among high school students to some degree increases in line with higher proportion of municipal residents with tertiary education. Although there are several plausible reasons why living in a high socioeconomic environment with a high proportion of well-educated inhabitants positively affects young people's health and well-being, the findings in the present study are contrary.

In our study, the Norwegian municipalities with high proportions of well-educated people were typically characterized by high household incomes and low disability pension rates, which have been used in comparison with lower educated municipalities as proxy measures of local economic and neighborhood conditions in multilevel analyses of health status [42,97]. It is argued in the literature that education creates social benefits to society beyond private returns through, among other factors, civic participation, enhanced political behavior, and lower crime rates [75,76]. In accordance with these assumptions, we found that the education level in a municipality was positively associated with the voting rate, whereas it was inversely associated with the reported crime rate. Thus, the education level in a municipality might be a contextual indicator not merely of the local economic conditions, but also of social determinants of mental health such as level of social support and community social capital. Despite that, we found a positive association between municipality education level and adolescents' psychological distress scores. This may indicate that contextual domains typically related to community infrastructure and material living conditions are either not of crucial relevance when considering differences in adolescents' psychological distress between Norwegian municipalities, or that there are psychological mechanisms that mask or balance out the beneficial effects stemming from structural and materialistic factors. Elgar et al. [37] indicated in their study of 1,371 adolescents in seven European countries that socioeconomic inequality in health is more closely related to psychosocial processes than to material inequality. Living in a community with a high share of the adolescents in socioeconomic advantageous families may, however, trigger some types of psychological stress. For example, expectations can be an important source of stress [98–102], and adolescents living in well-educated communities might perceive that the people they are surrounded by demand and expect much from them, including their parents, other family members, neighbors, classmates, teachers, training partners, and team leaders, as well as society as a whole. Furthermore, during adolescence the sense of social position is developing, and this psychological mechanism might also influence adolescents' health and well-being [35,72,103]; perhaps this source of stress is particularly prevalent in communities with high degrees of symbolic capital [31]. The evolutionarily based social rank theory (SRT) accounts for the inferiority and submissiveness that is typical in depression [65,71,104]. It might be that well-educated communities are characterized by underlying factors that increase the pursuit of social status and the chances of adolescents' feelings of inferiority. Societies that accentuate individual competitiveness will be particularly vulnerable to the emotionally disturbing effects of social comparison [65].

Furthermore, there are several other underlying contextual features that potentially explain why the average depression and anxiety scores among our study sample rise with increased municipal education level. It should be taken into consideration that the immigration rate was nearly three times higher in the most educated municipalities compared with the least educated municipalities. Moreover, major differences were observed regarding the municipalities' per capita numbers. The least educated municipalities had the smallest population size; in contrast, municipalities where the 60th percentile of adult inhabitants had a tertiary education had the highest populations. Differences in psychological distress between settlement types have been shown in several other studies, but the results are mixed. Rural and urban communities differ in environmental factors such as culture, socioeconomics, and access to healthy diets

and health care that may influence adolescents' functioning and well-being. In many Western countries, urban residents have a higher likelihood of suffering from psychological distress than rural residents [105]. However, across the degree of centrality, small geographical variations in anxiety and depressive symptoms were reported in a previous study from Norway [106].

Although we found an effect of education level at the municipal level, it is important to note that we found small variations between municipalities in adolescents' depressive and anxiety symptom scores which is in line with previous Nordic population studies exploring geographical variations and rural-urban differences in mental health [106–108], This might be because Norway has a high-level welfare state which benefits all Norwegians no matter where in the country they live. Norway is considered an egalitarian welfare country with generally low income inequality (e.g., the Gini coefficient was 0.26 in 2017, compared to a Gini coefficient of 0.32 across OECD countries [89] and it imposes national regulation on social and health services. The Nordic countries are leaders in promoting health through public policy action [95]. In 2018, 92.3% of high school students attended public schools [109].

## Income inequality

The second research question in this study was whether the level of income inequality in a municipality affects the psychological symptom load among students. In line with other population-based studies [50,110,111], our findings linked income inequality with poor mental health outcomes. According to Wilkinson, Pickett [111], unequal societies become dominated by status competition and class differentiation, and consequently they experience more health disadvantages. The prominent and well-accepted phenomenon of income inequality/health association is explained by hypotheses operating at different ecological levels, from the individual up to the national [48]. Rising income inequality in a society can lead to a weakening of collective social capital and social integration, thereby affecting the residents' health [112,113]. A second possibility, which overlaps with the social capital mechanism, is that income inequality reinforces the negative health effects of social comparison through a more prominent status hierarchy in the local community [111,114]. According to the status anxiety hypothesis, feelings of social defeat and inferiority produce stress reactions that lead to poorer health [115]. However, the contextual mechanism behind the income inequality/health association phenomena could also be interpreted without psychological explanations, but rather based on distribution of material resources in a society. The neo-materialist hypothesis argues that material conditions per se and public and social infrastructure influence individual well-being and health [44,116–119]. A review study analyzing data from 27 European countries [120] found the most support for the status anxiety hypothesis and the social capital hypothesis, and the author suggests that the mechanisms through which income inequality influences mental well-being vary depending on the wealth of the country. Similarly, a review of the inequality-depression relationship by Patel et al. [48] suggests that the main mechanisms at the local level are both the status anxiety hypothesis and the social capital hypothesis. At the individual level, the inequality-depression association is likely to be primarily mediated through psychological stress [48,49]. The heterogeneity of study findings across populations on the effect of income inequality on depressive symptoms reflects the complexity of mechanisms and pathways [48]. The size and geographical unit of analysis matters when considering the link between income inequality and mental health [121] because different geographical levels have different meanings and do not relate to the same contextual characteristics [122]. A study from The Netherlands, where income inequality is almost as low as in Norway, found an association between municipal income inequality and psychological distress (measured with the Kessler

Psychological Distress Scale) in the adult population [121]. Our study found that estimated income inequality had more effect on depressive symptoms than on anxiety symptoms.

Notably, in the present study, well-educated municipalities were positively associated with higher income inequality. This can be seen from an urban-rural perspective. Both level of education and average income characterize municipalities within a relatively large city. In geographic areas where there are many knowledge-based jobs, many well-educated people will reside, and this is reflected in the average income of the citizens. However, there is still a need for low-income jobs in these areas, which apparently creates a connection between education level and income inequality in Norwegian municipalities.

## Family affluence, and the case of medium affluent students

The third and final research question we raise in the present study is whether the degree of association between socioeconomic municipal characteristics and adolescents' psychological distress depends on family affluence level. We found that the psychological disadvantages resulting from increasing municipal education levels and income inequalities were most significant among adolescents living in families at the medium affluence level. Our parametric estimates suggest that as the municipal education level and income inequalities increase, the depressive and anxiety symptom mean scores in families of low and medium affluence converge, while the differences between those in families of medium and high affluence are slightly increased. It is somewhat puzzling why the students in families of medium affluence (measured at the national level in Norway) seem to be most affected by an increase in municipal education level. This may partly be explained by the fact that students in families of medium affluence residing in the least educated municipalities have proportionally more peers at lower family affluence levels than themselves compared to students of medium affluence living in the most highly educated municipalities in Norway. Consequently, students in families of medium affluence may have more to lose than affluent students in terms of socioeconomic rank by residing in highly educated municipalities. A more open question asks why the difference in average psychological symptom scores between students of low and medium affluence gradually becomes smaller with increasing socioeconomic level. One speculation is that students of medium affluence might be (somewhat) more concerned about their social status than students of low affluence. In a study of adolescents aged 14–17 years from six European cities, the socioeconomic inequalities in adolescent health were more closely related to adolescents' perceptions of relative family SEP than to objective indicators of material family affluence [123]. Thus, our findings give some support to the assumption that late adolescents' psychological symptom loads depend not only on their families' affluence levels, but also on the socioeconomic position that level of affluence confers in the social hierarchy (see Subramanian, Kawachi [124] for a brief review of linking income inequality, relative income, and relative rank to health).

In general, our likelihood estimates show the highest psychological distress among students in families with the lowest family affluence, while the most affluent families had the fewest students suffering from anxiety and depressive symptoms. By using the FAS, Elgar et al. [37] demonstrated socioeconomic differences in adolescents' mental health in teenagers aged 11–15 years from high-income countries. Our study indicates that the late period of adolescence (i.e., 16–18 years) is not exempt from a socioeconomic family gradient of mental health. Given that people use conventional socioeconomic criteria to assign themselves subjective status [125,126], we may assume that family affluence is involved in adolescents' assessments of their subjective SEP relative to their peers. Adolescents who perceive themselves to be at a lower level in the socioeconomic hierarchy compared to their peers are at higher risk of poor well-being and health outcomes [35,72,127].

## Strengths and limitations

A main strength of the study is its relatively large and nationally representative sample, with explanatory factors at the individual and family levels linked to population-based municipal socioeconomic factors from national administrative databases. Moreover, family affluence level and adolescents' symptoms of depression were evaluated in a standardized manner using validated measures [40,84,85]. There are, however, several limitations of this study that should be noted. First, the use of a cross-sectional design limits the interpretation of our findings as we cannot draw inferences regarding the direction of the observed associations or the risk factor status of explanatory variables. Nor can our results reveal what may be the underlying causes of the associations we found. Second, although overall response rates were relatively high, approximately 30% of students were absent from the surveys, which inevitably poses the possibility of non-response bias due to illness and truancy. Third, while the items from the Depressive Mood Inventory have been validated in clinical studies [84,128], the present combination of anxiety items has not been validated. Thus, the reliability of the anxiety measure is uncertain, and the use of exclusively validated instruments would have strengthened the study findings. Fourth, the various high schools and municipalities included in the study sample differ in each survey/study year. Fifth, using municipality as an area-level when investigating how community environments may be related to health outcomes poses methodological challenges [59]. Other geographical areas and social contexts (e.g., neighborhoods, school districts, peer groups, and other types of communities that facilitate social interaction) may be more relevant than municipalities for adolescents' well-being and mental health outcomes. Moreover, the socio-demographic distribution of the population varies between Norwegian municipalities, and it is important to take this into account when interpreting the results of this study. However, presenting and interpreting estimates of effect measures for secondary risk factors (confounders and modifiers of the exposure effect measure) from a single statistical model may lead to several interpretative difficulties and may be misleading [129]. Sixth and finally, an ideal study examining contextual effects on various health outcomes should include explanatory variables at multiple levels and allow the levels (i.e., contexts) to change over time. Family background, neighborhood of residence, and schools the children attended before high school are three highly important levels, and adjustment for them would constitute a strong advancement of the study. Ideally, this should be performed using a multiple-membership, cross-classified, multilevel analysis that allows the levels (municipality, neighborhood, school, and families) to change over time. However, our data do not include information about school, neighborhood, or particular household the child attended before entering secondary school, and thus this analytic framework cannot be used in this case.

## Interpretations and conclusion

Living in a community with a high proportion of well-educated people is associated with several benefits. One possible benefit is the opportunity to learn and imitate healthy behaviors. Another benefit may be the sharing and receiving of social capital as well as the materialistic facilities in these types of communities. Despite these potential benefits of living in a well-educated society, our study demonstrates that there must be other factors at play which apparently are even more important for psychological well-being in the high-income and egalitarian welfare state of Norway with its national regulation of social benefits and health services. Thus, there may be psychological mechanisms associated with well-educated communities that have negative effects on adolescents' mental health. The Norwegian municipalities with high proportions of well-educated people were typically characterized by an increase in income inequality compared to less educated municipalities. Living in competitive-oriented

environments with high degrees of demands and expectations in several social arenas (such as at home, school, and sports activities, and when socializing with friends both in the virtual world and when physically present) might generate "status anxiety" and daily stress. We suggest that this type of disadvantageous environment is amplified by increased education level and income inequality in the Norwegian municipalities. Although we identified a modest degree of association between adolescents' socioeconomic circumstances and psychological distress in Norway, the findings have potentially important implications for population health and society at large which should be considered when developing community planning and policy in high-income countries facing social changes and within-region inequalities in socio-economic status.

## Supporting information

**S1 Fig. Predictive margins of high, medium and low affluence high school students predicting probability (Pr) of moderate to high depressive and anxiety symptoms complaints by municipal education level.**
(TIF)

**S2 Fig. Predictive margins of high, medium and low affluence high school students predicting probability (Pr) of moderate to high depressive and anxiety symptoms complaints by municipal income inequality (Gini coefficient).**
(TIF)

**S1 Table. Odds ratios (ORs) of moderate-to-high psychological symptoms by use of multi-level logistic regression models.**
(DOCX)

**S2 Table. The impact of family affluence, municipal education level and their interactions\* for risk of moderate to high levels of psychological distress, depressive and anxiety symptoms among high school students in Norway.**
(DOCX)

**S3 Table. The impact of family affluence, municipal income inequality (Gini-coefficient) and their interactions\* for risk of moderate to high levels of psychological distress, depressive and anxiety symptoms among high school students in Norway.**
(DOCX)

## Acknowledgments

The Ungdata surveys were conducted by the Norwegian Social Research (NOVA) institute in cooperation with regional centers for drug rehabilitation (KoRus). We wish to thank them for their cooperation and for conducting the data collection.

## Author Contributions

**Conceptualization:** Tommy Haugan, Sally Muggleton, Arnhild Myhr.

**Formal analysis:** Arnhild Myhr.

**Investigation:** Tommy Haugan, Arnhild Myhr.

**Methodology:** Tommy Haugan, Arnhild Myhr.

**Project administration:** Tommy Haugan.

**Validation:** Tommy Haugan, Arnhild Myhr.

**Writing – original draft:** Tommy Haugan.

**Writing – review & editing:** Sally Muggleton, Arnhild Myhr.

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
