## [Decision Letter · Decision Letter 0]

19 Apr 2021

PONE-D-21-02793

Psychological distress in late adolescence: The role of inequalities in family affluence and municipal socioeconomic characteristics in Norway

PLOS ONE

Dear Dr. Haugan,

Thank you for submitting your manuscript to PLOS ONE. After careful consideration, we feel that it has merit but does not fully meet PLOS ONE’s publication criteria as it currently stands. Therefore, we invite you to submit a revised version of the manuscript that addresses the points raised during the review process.

We look forward to receiving your revised manuscript.

Kind regards,

Xiaozhao Yousef Yang, Ph.D.

Academic Editor

PLOS ONE

Journal Requirements:

Reviewers' comments:

Reviewer's Responses to Questions

**Comments to the Author**

1. Is the manuscript technically sound, and do the data support the conclusions?

Reviewer #1: Yes

2. Has the statistical analysis been performed appropriately and rigorously? 

Reviewer #1: Yes

3. Have the authors made all data underlying the findings in their manuscript fully available?

Reviewer #1: Yes

4. Is the manuscript presented in an intelligible fashion and written in standard English?

Reviewer #1: Yes

5. Review Comments to the Author

Reviewer #1: I think this manuscript titled: "Psychological distress in late adolescence: The role of inequalities in family affluence and municipal socioeconomic characteristics in Norway" is of great interest and, in general, is well organized and well written. The topic is exact and relevant in the actual situation where mental health is more than ever valued. Besides, it brings to the reader new insights about the variables to consider in terms of youth, mental health, and social contexts and their associated features (family, municipalities, ...). Although focused on psychological distress in adolescence, it highlights several psycho-social aspects underlying a new vision and understanding about youth mental health. I think this psycho-social focus on mental health is one of the significant innovative contributions of the study. The outcome is a study that contributes to deep analysis and understanding of a psychological issue but going out from a psychologically perspective and thus becoming a comprehensive tool for researchers and practitioners. Also of relevance is the large and nationally representative sample used in the study.

Still, I have a few comments on strengths, and also suggestions about possible changes to improve the article. Following the structure of the manuscript, they are:

1. INTRODUCTION

Very well organized and clear, straight to the point.

The authors did a good summary of fundamental literature to support the study's aim and the problem. Although being targeted in Norway, the information is sufficiently valuable and adequate to understand other countries through this particular social organization's lens. Even related to just one country, the N of participants is significantly large, making this a robust study with broad implications and applications.

The introduction in subsections makes it easy to follow and incorporate main conceptual aspects/issues and variables to consider.

Study aims and research questions are clear and emerging from the introduction's contents.

2. METHODS

- Sample and data collection

The authors present a lot of pertinent information about participants and the data collection. According to this, some more aspects could be pointed out in a more explicitly way (e.g using subsections/subtitles), for example, if there were any eligibility criteria to participate or if there was a differentiation between the clinical and normative population, or if both were integrated into the study or not. The sentence (see lines from 216-218): "We found no significant differences between the study sample and excluded individuals in terms of mean symptom scores of psychological distress (1.91 vs. 1.91), depression (2.16 vs. 2.15), or anxiety (1.55 vs. 1.55)" is not clear, please add the word "respectively" at the end of the sentence. Besides, giving some information/characteristics about excluded individuals could help understand reported values and compare excluded participants and the study sample.

- Measures

This section requires a more attentive organization in terms of titles and subtitles.

Some doubts emerged: are the titles of "Measures" and "Individual and family level" and "Municipality level" at the same level, or all these points are about measures and so "Individual and family level" and "Municipality level" should be inside the section "Measures" as are the first subtitle: The outcome variables: psychological distress, symptoms of depression and anxiety"? Please review this organization of contents inside the subsection "Measures", even if authors considered different measures under each level.

Besides, when reporting specific scales, I suggest informing the particular version used because some scales have several different versions (with, for example, different number of items); and/or if there were some changes in the original ones to serve specific purposes in this study.

This means that more information on measurement instruments should be provided.

- Statistical approach

I think the proposed statistical analysis is according to the study's aims and data.

Besides this section, the authors also report some other specific analysis inside other parts of the manuscript (e.g., family and municipality levels have references to particular statistics). I suggest the authors write in this section of the "statistical approach" that some other information on specific statistics will be (or have already been) shown in specific other parts of the article.

3. RESULTS

I think this is a strong part of the manuscript.

According to study aims and methods, the authors obtained several types of results. Tables and Figures are very elucidative, clear and readable, aggregating significant results and improving readers' facility to follow all the information.

4. DISCUSSION

In the discussion, the authors summarize the main findings, showing how the results support the conclusions. The authors highlighted the leading and innovative findings produced by the study in an excellent way, supporting and interpreting them clearly, through well-supported literature and previous research. Although this study focus in a specific country, the new and unexpected results and conclusions can inspire and support other studies on youth mental health in general. Still, I ask if authors could explicitly re-organize some of the research questions' findings. These questions defined at the beginning (in the introduction) should be explicitly again in the discussion in order to benefit the manuscript (just as a suggestion).

The authors also report the study's strengths and limitations, and they also give suggestions to overcome some of these limitations in future studies. this is a very relevant part in the discussion.

The list of references is complete and diverse, with very recent references supporting the topics discussed along with the article.

In general, I think this is a relevant study demonstrating significant advances in adolescent mental health.

I hope authors will reflect on comments made and improve the manuscript following suggestions on few aspects.

6. PLOS authors have the option to publish the peer review history of their article (what does this mean?). If published, this will include your full peer review and any attached files.

Reviewer #1: No

---

## [Author Response · Author response to Decision Letter 0]

27 May 2021

Dear editor and reviewer,

In the revised manuscript we have endeavored to meet PLOS ONE’s style requirements and made some changes based on comments from reviewer.

Response to the reviewer.

Thank you for constructive suggestions for improvements in the method sections. In the revised manuscript under “Methods” we have made the following revisions: 

1. Sample and data collection

- renamed the "sample and data collection" into "Design and data sources" and included the following subtitles "The ungdata survey". "Municiplaity state reporting (KOSTRA)" and "study population", please see line 183-235. 

- rewritten the following sentences (line 216-219): "We found no statistical difference between included and excluded individuals for mean symptom scores of psychological distress (1.91 vs. 1.91), depression (2.16 vs. 2.15), or anxiety (1.55 vs. 1.55). There was a noted higher share of first-year students (54 % versus 45 %, p<0.001) for excluded compared to included groups.

2. Measures 

- Rewritten the section into "assessment of variables" with the following subtitles: " Psychological distress, and depressive and anxiety symptoms", " Family affluence as a proxy for socioeconomic position (SEP)", " Municipal sociodemographic characteristics " and "Covariates"

- Revised the sections “Psychological distress, and depressive and anxiety symptoms” and “Socioeconomic position” – with clearer and more information of the particular scales used. 

3. Statistical approach 

- We included the following sentence in the first part of the statistical approach section (line 321-323): "First, descriptive analyses of percentages on individual variables and municipal sociodemographic variables, by municipal education level were tested by chi square tests and analysis of variance (ANOVA)".

We are also very grateful for the reviewer's advice to reorganize parts of the “Discussion” chapter. In the revised version of the manuscript, we have highlighted our three research questions in separate sections. In each of these sections, we first answer the respective research question by stating our related main finding.

Data Availability statement

The data and materials from the Ungdata-surveys are closed and stored in a national database administered by Norwegian Social Research (NOVA). The present study and analysis of the Ungdata were approved by the Norwegian Centre for Research Data (NSD). Norwegian legislation prohibits deposition of these data to open archives. The data are freely available for research purposes upon application. Details about the application process to NSD can be found at: https://nsd.no/nsddata/serier/ungdata_eng.html.

Yours sincerely,

Tommy Haugan

---

## [Editor Report · Decision Letter 1]

18 Jun 2021

Psychological distress in late adolescence: The role of inequalities in family affluence and municipal socioeconomic characteristics in Norway

PONE-D-21-02793R1

Dear Dr. Haugan,

We’re pleased to inform you that your manuscript has been judged scientifically suitable for publication and will be formally accepted for publication once it meets all outstanding technical requirements.

Kind regards,

Xiaozhao Yousef Yang, Ph.D.

Academic Editor

PLOS ONE
---

## [Editor Report · Acceptance letter]

23 Jun 2021

PONE-D-21-02793R1 

Psychological distress in late adolescence: The role of inequalities in family affluence and municipal socioeconomic characteristics in Norway 

Dear Dr. Haugan:

I'm pleased to inform you that your manuscript has been deemed suitable for publication in PLOS ONE. Congratulations! Your manuscript is now with our production department. 

Kind regards, 

on behalf of

Dr. Xiaozhao Yousef Yang 

Academic Editor

PLOS ONE